# Dynamic Compressive Mechanical Properties of UR50 Ultra-Early-Strength Cement-Based Concrete Material under High Strain Rate on SHPB Test

**DOI:** 10.3390/ma15176154

**Published:** 2022-09-05

**Authors:** Wei Wang, Zhonghao Zhang, Qing Huo, Xiaodong Song, Jianchao Yang, Xiaofeng Wang, Jianhui Wang, Xing Wang

**Affiliations:** 1Key Laboratory of Impact and Safety Engineering, Ningbo University, Ministry of Education, Ningbo 315211, China; 2Institute of Advance Energy Storage Technology and Equipment, Ningbo University, Ningbo 315211, China; 3Institute of Defence Engineering AMS, PLA, Luoyang 471023, China

**Keywords:** ultra-early-strength concrete, split Hopkinson pressure bar (SHPB), impact mechanics, dynamic response, strain rate effect

## Abstract

UR50 ultra-early-strength cement-based self-compacting high-strength material is a special cement-based material. Compared with traditional high-strength concrete, its ultra-high strength, ultra-high toughness, ultra-impact resistance, and ultra-high durability have received great attention in the field of protection engineering, but the dynamic mechanical properties of impact compression at high strain rates are not well known, and the dynamic compressive properties of materials are the basis for related numerical simulation studies. In order to study its dynamic compressive mechanical properties, three sets of specimens with a size of Φ100 × 50 mm were designed and produced, and a large-diameter split Hopkinson pressure bar (SHPB) with a diameter of 100 mm was used to carry out impact tests at different speeds. The specimens were mainly brittle failures. With the increase in impact speed, the failure mode of the specimens gradually transits from larger fragments to small fragments and a large amount of powder. The experimental results show that the ultra-early-strength cement-based material has a greater impact compression brittleness, and overall rupture occurs at low strain rates. Its dynamic compressive strength increases with the increase of strain rates and has an obvious strain rate strengthening effect. According to the test results, the relationship curve between the dynamic enhancement factor and the strain rate is fitted. As the impact speed increases, the peak stress rises, the energy absorption density increases, and its growth rate accelerates. Afterward, based on the stress–strain curve, the damage variables under different strain rates were fitted, and the results show that the increase of strain rate has a hindering effect on the increase of damage variables and the increase rate.

## 1. Introduction

With the rapid development of precision-guided weapons, facilities in the field of protection engineering are facing a serious threat of weapons, “accurately and aggressively”. Therefore, the design and construction of military protection engineering must consider the dynamic response characteristics and parameters of concrete. Ultra-early-strength cement-based materials, because of their excellent performance, have attracted great attention in airport engineering repairs and protection engineering. Ultra-early-strength cement-based materials are usually subjected to strong dynamic load impact compressions during service, like ammunition penetration and explosion. Under blast loads, concrete structures can crater on the surface, peel off the concrete on the back, or even crack [1,2,3]. The dynamic compression performance of the material plays an important role in studying the dynamic mechanical response process of materials under penetration and explosion. Therefore, investigating the performance of ultra-early-strength cement base under dynamic load and establishing a dynamic stress–strain curve has important military value and engineering significance.

At present, although domestic and foreign researchers have carried out many experimental studies on concrete materials, there are few studies on the dynamic mechanical properties of UR50 ultra-early-strength cement materials under high dynamic loads. Pang et al. [4] studied activated fly ash concrete under high strain rates with an SHPB device. Its dynamic strength is affected by both temperature and the water–binder ratio. At room temperature, the dynamic strength is directly proportional to the water–binder ratio, but there is a negative correlation at high temperatures. Hu et al. [5] applied an SHPB device to study the spalling strength and strain rate effects of concrete materials finding that spalling strength and strain rate are positively correlated. Zhu et al. [6] conducted a numerical simulation study on the SHPB test of cement materials. The strain value and strain rate of cement materials will be strongly affected by the amplitude of the incident wave. Bragov et al. [7] investigated the mechanical properties of fine-grained concrete at high strain rates with an SHPB device. Levi-Hevroni et al. [8] used the SHPB test to explore the dynamic reinforcement factor and tensile strength of concrete first, then applied the test data to calibrate the three concrete material model parameters in LS-DYNA. Zhang et al. [9] studied the influence of the specimen shape on the dynamic increase factor under a high strain rate based on the SHPB test device. The results showed that the dynamic enhancement factor of the tubular specimen was lower than that of the cube. Wang et al. [10] studied the physical mechanism of the static–dynamic composite multiaxial strength of concrete under the premise of considering the cohesive and frictional strength. Gu et al. [11] introduced the theory of non-local circumferential dynamics to the analysis of dispersion and impact failure of elastic waves in SHPB tests and verified the feasibility of the analysis by experiments and numerical simulations. Hassan and Wille [12] studied the dynamic mechanical properties of ultra-high-performance concrete (UHPC) at high strain rates based on the SHPB test device. Erzar and Forquin [13] studied the effects of aggregate and free water on the mechanical properties of concrete materials under high strain rates through experiments and numerical simulations. 

With the development of research, some scholars have begun to explore the dynamic mechanical properties of new composite concrete materials. Wang et al. [14] studied the failure mode and energy absorption mechanism of autoclaved aerated concrete under low-velocity impact. Kang [15] explored the mechanical behavior of foam-insulated concrete sandwich panels under uniform loads through experiments and numerical simulations. Wang et al. [16] studied the mechanical properties of lightweight aggregate foam concrete at different compression rates and found that the compressive strength is directly proportional to the density of the foam concrete. Shafigh et al. [17] used oil palm shells to prepare lightweight concrete and tested the compressive strength under different curing times. Cao et al. [18] discussed the influence of specimen size on the dynamic compression performance of fiber-reinforced reactive powder concrete at high strain rates. Xiong et al. [19] studied the dynamic mechanical properties of Carbon Fiber Reinforced Polymer (CFRP) confined concrete at a high strain rate based on an SHPB test device with a diameter of 155 mm. The results show that CFRP-confined concrete is not sensitive to the strain rate effect. Liu et al. [20] used a separate SHPB device with a diameter of 100 mm to study the influence of the content of the redispersible polymer emulsion powder on the dynamic mechanical properties of Carbon Fiber Reinforced Polymer Concrete (CFRPC). The dynamic compressive strength of carbon fiber composites increases firstly and then decreases with the increase of polymer content. Wei et al. [21] used SHPB to study the dynamic response of a ceramic shell for titanium investment casting under high strain rates. Ceramic shells are highly sensitive to the strain rate effect, and the path of crack propagation is different under quasi-static and high strain rate loads. Sun et al. [22] used a 75 mm diameter SHPB to study the dynamic mechanical properties of steel fiber-reinforced concrete at different strain rates and steel fiber content. As the strain rate or steel fiber content increases, the ductility, strength, and toughness will increase. Scott et al. [23] established a constitutive model of the dynamic response characteristics of concrete materials based on a large amount of experimental data. Georgin and Reynouard [24] established a viscoelastic model of the strain rate effect and applied it in a numerical simulation. Xiuli et al. [25] used SHPB to conduct a uniaxial dynamic compression test of concrete materials, and based on this, established a non-linear uniaxial dynamic strength criterion for concrete materials. The temperature has a certain influence on the dynamic properties of materials. Aiming at the dynamic mechanical properties of concrete specimens under the coupled action of high temperature and impact, Huo et al. [26] used SHPB to carry out the impact resistance test of concrete-filled steel tube specimens at a high temperature of 400 °C, and the results showed that the restraint of steel tubes improved the impact resistance of concrete specimens.

With the development of material technology, convenient, efficient, and excellent-performance concrete materials have begun to attract people’s attention. Ultra-early-strength cement-based self-compacting high-strength material has the advantages of good fluidity, ultra-fast hardening, and high strength. It is easy to use and can be mechanically stirred by adding water on site. It has attracted the attention of the field of protection engineering. However, there is little research on ultra-early-strength cement-based self-compacting high-strength materials. In contrast to previous studies, the dynamic compressive properties of the material were not investigated. In order to study the dynamic mechanical response process of materials under the action of penetrating explosions and lay the foundation for numerical simulation research, this paper conducts related research. In this paper, a separate SHPB test device with a diameter of 100 mm and an ultra-early-strength cement-based self-compacting high-strength material with a product code of “UR50” were used to conduct impact tests at different loading speeds to explore the dynamic mechanical properties of this new type of concrete material and provide data support for studying the impact resistance and numerical simulation of UR50 ultra-early-strength cement-based materials.

## 2. UR50 Ultra-Early-Strength Concrete Material

UR50 ultra-early-strength cement-based self-compacting high-strength material is a special cement-based material. It is a pre-dry mixed powder composed of aggregate, cement, functional mineral powder, nano filler, specially modified additives, and special steel fiber. The maximum particle size of the aggregate is less than 5 mm. The product is processed and mixed in the factory and packed in bags, and the shelf life is about 6 months. After adding water and mixing on site, it has good fluidity, ultra-high strength, ultra-high toughness, ultra-impact resistance, and ultra-high durability. Wang et al. [27] found that the microstructure was greatly improved compared to conventional high-strength concrete, the pores were eliminated, and the nano-microstructure was strengthened. The design of dry mixing and pre-dispersed low-proportion components greatly improves the strength and durability of concrete, and this method makes the microstructure of concrete denser.

UR50 ultra-early-strength cement-based self-compacting high-strength material is convenient to use, and after adding water and mechanical mixing, concrete materials with excellent fluidity, super-fast hardening, and high strength can be obtained immediately. The amount of water added is 9.3 ± 0.5% of the weight of the dry powder. The slump extension of the mixed concrete can reach 770–830 mm, and the pouring can be self-compacting without vibration.

The early strength of UR50 ultra-early-strength cement-based materials has developed rapidly, and the compressive strength can reach 50 MPa in 2 h, 70 MPa in 24 h, and even the later compressive strength exceeds 80 MPa. The compressive strength changes with time, as shown in Table 1.

## 3. Dynamic Test with SHPB

### 3.1. Experimental Specimen

The size of the experimental specimen was designed as Φ100 × 50 mm in order to avoid the side wall effect caused by the specimen mold during the curing process and the discreteness of the specimen itself caused by the inconsistent curing and vibrating conditions. It was ensured that the test specimen met the relevant international specifications [28]. Adopting the method of pouring ultra-early-strength cement-based material panels and after curing for 28 days, the core was taken from the panel by using the coring machine to take it out of the poured ultra-early-strength cement-based material panel. The thickness of the ultra-early-strength cement-based material plate was about 60 mm, and the plate was vibrated evenly with a vibrating table and then placed in the pool for curing for a scheduled time and then taken out. The inner diameter of the core bit was 100 mm. After the core is taken, the upper and below faces were ground to a thickness of 50 mm with a grinder and polished to make a test specimen with a size of Φ100 × 50 mm. There were three sets of test specimens and each group had three specimens. The finished test specimens are shown in Figure 1.

### 3.2. Experimental Device

The impact compression test equipment was a Φ100 mm SHPB of the Engineering Protection Research Institute. The test device is mainly composed of an operating console, launching device, impact bar (bullet), speed measuring device, incident bar, transmission bar, support, absorption bar, buffer device, measuring device, etc. The pressure bar is made of high-strength spring steel, as shown in Figure 2a. In order to effectively eliminate the influence of friction on the support, rolling bearings were installed at the support of the device. The length of the incident bar was 4500 mm, and the aspect ratio of the transmission rod was 25, which meets the relevant requirements [29]. The length of the transmission bar was 2500 mm, and the length of the striker bar was 500 mm and 800 mm, respectively. The compressed gas pressure was controlled by the operating console to control the impact speed of the bullet. The impact speed was measured by the speed measuring device. Strain gauges were attached to the incident bar and transmission bar to measure the incident wave, reflected wave, and transmitted wave. The strain gauge was connected to the super dynamic strain gauge through the Huygens bridge, and after being amplified by the strain gauge, it was saved as a transient record. The original waveform was analyzed and processed by a self-compiled data processing program to obtain the stress–strain rate relationship, as shown in Figure 2b.

### 3.3. Experimental Design

In the experiment, the impact velocity of the bullet was altered by changing the driving air pressure so as to obtain the stress–strain curve of the material under different strain rates. A total of three kinds of loading speeds (impact speeds of 5 m/s, 10 m/s, and 15 m/s) for the SHPB impact compression experiments were carried out. Three repeated experiments were carried out for each speed state, and the average curve of these three experiments was taken as the stress–strain curve of the material under the strain rate. The experiment process of ultra-early-strength cement-based material is shown in Figure 3. The experiment loading speed and test specimen number are shown in Table 2.

### 3.4. Calibration of SHPB

The basic principle of SHPB is to decouple the wave propagation effect and the strain rate effect of the material and then separate the strain rate effect of the material. Figure 4a shows the electrical signal curve collected by the data acquisition system, which represents the curve of voltage change over time. The recorded data can be restored to stress and strain curves by data processing software. Figure 4b is the three-wave diagram of the incident wave, reflected wave, and transmitted wave obtained from the experiment. The comparison wave is the incident wave + the reflected wave. It can be seen from the figure that the transmitted wave is in good agreement with the comparison wave, which meets the criterion of $\varepsilon_{t}\left(t\right)=\varepsilon_{r}\left(t\right)+\varepsilon_{i}\left(t\right)$, proving that the test data are valid and can be used for analysis.

In order to make the stress pulse have enough time to reflect back and forth before the failure of the ultra-early-strength cement-based material specimen to obtain a uniform distribution of stress in the specimen, wave shapers were installed on the impacted end of the incident bar, which can eliminate the overshoot and wave oscillation of the stress wave caused by the dispersion effect of the large-size SHPB device, and it is helpful to obtain the true response characteristics of the material. In the experiment, under different loading conditions, different shapers were selected. Among them, the shaper used under low strain rate loading conditions (impact velocity 5 m/s) was a Φ16 × 0.5 mm copper sheet. Under the condition of medium strain rate loading (impact velocity 10 m/s), a Φ30 × 2 mm copper sheet shaper was used. Under high strain rate loading conditions (impact velocity 15 m/s), the shaper is a Φ30 × 4 mm copper sheet. By choosing a proper shaper, the dispersion effect can be eliminated effectively, the change in the waveform when the wave propagates in the waveguide bar is reduced, and the accuracy of the experiment is improved. 

Assuming that the strain rate is constant during the loading process, the average strain rate during loading can be determined by Formula (1):(1)$$\dot{\varepsilon }=\varepsilon /t$$

If t=100 μs and the failure strain ε=8500 με, it can be determined that the highest strain rate that can satisfy the stress uniformity is 85 s^−1^. It is particularly pointed out that the strain rate of the specimen is not constant during the experiment. During the data processing, the average strain rate of the loading stage before the failure of the specimen is taken as the average strain rate.

## 4. Test Results

### 4.1. Average Strain Rate of 7.5 s^−1^

Table 3 shows the macro morphology of UR50 ultra-early-strength cement-based specimens after impact compression at different stages when the impact velocity is 5 m/s (the average strain rate is 7.5 s^−1^) in three repeated experiments. It can be seen from the table that at an impact velocity of 5 m/s, the specimens were damaged under the impact compression stress wave. The specimen ruptured into a number of larger fragments, indicating that the ultra-early-strength cement-based material has a greater impact compression brittleness. Under the low strain rate, the overall fracture will occur, but under the same strain rate conditions, ordinary concrete will generally not fail. The stress–strain curve and the average stress–strain curve obtained by repeating the experiment three times at an impact velocity of 5 m/s are shown in Figure 5.

### 4.2. Average Strain Rate of 15.3 s^−1^

The macro morphology of the ultra-early-strength cement-based material specimen after three repeated experiments, when the impact velocity is 10 m/s (the average strain rate is 15.3 s^−1^), can be seen in Table 4. It can be seen from the table that, at an impact velocity of 10 m/s, the specimen was severely damaged under the impact of the compression stress wave. Because of the short time of the load, cracks appeared throughout the whole specimen at the moment of impact and rapidly expanded until it broke into many smaller pieces. The stress–strain curve and the average stress–strain curve obtained by three repeated experiments are shown in Figure 6. The consistency of the repeated experiments is great.

### 4.3. Average Strain Rate of 23.5 s^−1^

The macro morphology of the ultra-early-strength cement-based material specimens in each stage of the three repeated experiments, when the impact velocity is 15 m/s (the average strain rate is 23.5 s^−1^), can be seen in Table 5. At an impact velocity of 15 m/s, the specimen was impacted within a very short time after the bullet was launched, accompanied by a loud noise. Afterward, it can be observed that the concrete specimens were a comminuted failure under the impact compression stress wave, and the specimens were broken into a large amount of powder and small pieces. Figure 7 shows the stress–strain curve and the average stress–strain curve obtained from three repeated experiments at an impact velocity of 15 m/s. It can be seen from the figure that the curve rises sharply at the beginning of the impact, and subsequently, the compressive strength of the specimen quickly reached the peak point due to its own strain rate sensitivity, and the second half of the curve began to gradually decrease after the end of the loading.

## 5. Discussion

### 5.1. Strain Rate Effect and Analysis of Compressive Strength

Figure 8 is a summary of the stress–strain curves of ultra-early-strength cement-based materials at different strain rates. The impact velocities are 5 m/s, 10 m/s, and 15 m/s, and the corresponding average strain rates are 7.5 s^−1^, 15.3 s^−1^, and 23.5 s^−1^, respectively. Experimental results show that ultra-early-strength cement-based materials are strain-rate sensitive materials, and the stress–strain curves of the materials at different strain rates are significantly different. When the strain rate is 7.5 s^−1^ (impact velocity 5 m/s), the stress–strain curve shows a yield platform, and it shows that under this loading condition, the ultra-early-strength cement-based material specimen enters an obvious yielding stage from the elastic stage.

After reaching the peak point, the specimen was damaged. The stress–strain curve cannot be unloaded to the zero point, indicating that during the loading process, damage evolution occurred inside the concrete, and the specimen produced plastic deformation.

When loading at 10 m/s and above, the stress strain curve of the ultra-early-strength cement-based material does not show an obvious yield platform, and it transitions directly from the elastic stage to the yield stage. An obvious strain softening phenomenon appears after the peak stress (the stress decreases with increasing strain), indicating that the specimen still has the load-bearing capacity. The stress–strain curve at the strain software stage is no longer the mechanical response of the initial complete material, but the specimen still has a certain residual strength at this time. At this stage, the stress–strain curve still has engineering significance for analyzing the damage and destruction of concrete structures under the explosion and impact load, so the curve of the strain-softening stage is still retained in data processing.

The phenomenon that the dynamic compressive strength of concrete materials increases with the increase in strain rate has been confirmed by extensive experiments, but there is no unified conclusion on the mechanism of the strain rate effect of strength. The increase in the strain rate causes an increase in the strength of concrete materials, which is generally caused by the free water viscosity effect and the crack propagation inertia effect. The inertial effect of crack propagation is generally caused by the concrete matrix and aggregates. Considering that the ultra-early-strength cement-based materials do not contain aggregates, which is different from ordinary concrete, the main reason that the dynamic compressive strength of ultra-early-strength cement-based materials increases with the increase of the strain rate may be the effect of free water viscosity inside the material.

The dynamic increase factor, DIF, the ratio of the dynamic strength to the static strength of concrete, is used to characterize the dynamic characteristics of brittle materials frequently. Table 6 shows the dynamic compressive strength of ultra-early-strength cement-based materials obtained by SHPB impact compression loading experiments at three different strain rates.

In this paper, the DIF model Formula (2) of the concrete under the one-dimensional stress state approved by the Euro-International Committee for Concrete (the CEB) is used to fit the experimental data in Table 6.
(2)$$DIF=\frac{f_{c}}{f_{co}}=\{\begin{array}{cc} {\left(\frac{\dot{\varepsilon }}{{\dot{\varepsilon }}_{s}}\right)}^{\alpha_{c}} & \dot{\varepsilon }<k\\ \beta_{c}{\left(\frac{\dot{\varepsilon }}{{\dot{\varepsilon }}_{s}}\right)}^{\gamma_{c}} & \dot{\varepsilon }\geq k \end{array}$$

In the formula, *f_c_* is the corresponding compressive strength (MPa) when the strain rate is $\dot{\varepsilon }$, *f_c_*_0_ is the static compressive strength (MPa), $\dot{\varepsilon }$ is the strain rate, ${\dot{\varepsilon }}_{s}$(${\dot{\varepsilon }}_{s}$ = 3 × 10^−5^ s^−1^) is the quasi-static strain rate, $\alpha_{c}$, $\beta_{c}$ and $\gamma_{c}$ are fitting parameters, and *k* is the critical strain rate. The fitting results are as follows:(3)$$DIF=\frac{f_{c}}{f_{co}}=0.0018{\left(\frac{\dot{\varepsilon }}{{\dot{\varepsilon }}_{s}}\right)}^{0.5211}\;7.5\;s^{-1}\leq \dot{\varepsilon }$$

Figure 9 shows the relationship between the dynamic enhancement factor of ultra-early-strength cement-based materials and the strain rate. It can be seen from the figure that the dynamic compressive strength of the material has a significant strain rate effect, and as the strain rate increases, the dynamic compressive strength increases significantly.

### 5.2. Energy Absorption Density Analysis

Toughness is the ability of a material or structure to absorb energy under a load until it fails. It not only depends on the bearing capacity but also on the deformation capacity [30]. The methods to determine the toughness index include the energy method, intensity method, energy ratio method, characteristic point method, etc. In this paper, the energy method was used, and the area enclosed under the stress–strain curve is used to represent the characterization method of absorbed energy. The energy absorption density can be calculated by Formula (4).
(4)ω=∫σ*(ε)dε

Figure 10 shows the energy absorption density curve at different loading speeds obtained by Formula (4). It can be seen from the figure that under different impact speeds (strain rates), the energy absorption density increases with the increase of strain, and as the impact velocity increases, the growth rate of the energy absorption density accelerates. Figure 10a shows the relationship between energy absorption density and strain when the impact velocity is 5 m/s. The average energy absorption density at the stress peak point is about 1.5 × 10^5^ J/m^3^, and the corresponding peak strain is 3100 με. When the impact velocity is 10 m/s, as shown in Figure 10b, the corresponding curve of each test piece is relatively concentrated, and the corresponding energy absorption density at the peak stress point is about 3 × 10^5^ J/m^3^.

Relative to the impact velocity of 5 m/s, the energy density value doubles but the peak strain hardly changes. When the impact velocity increases to 15 m/s, the corresponding curves of each test piece are scattered slightly. As shown in Figure 10c, the curve becomes steep, compared with the low-speed impact, the corresponding energy absorption density at the peak stress point is about 2.3 × 10^5^ J/m^3^, and the corresponding peak strain is about 1500 με. According to the energy density absorption value, peak stress, and corresponding peak strain of the specimen under different impact speeds, it can be concluded that, as the impact velocity increases, the peak stress rises, the energy absorption density increases and its growth rate accelerates, as shown in Figure 10d. The peak strain at an impact velocity of 15 m/s is lower than that of low-speed impact (5 m/s and 10 m/s).

### 5.3. Damage Evolution Process Analysis

Under the impact load, cracks will appear inside the specimen, and the cracks will gradually propagate from the inside to the outside. When the cracks propagate to the boundary of the specimen, they will cause penetration and breakage. In this section, from the perspective of continuum mechanics, the region containing many scattered micro-cracks is regarded as a local uniform field, the overall effect of the crack in the field is considered, and the damage state of the uniform field is described by defining an irreversible related field variable, which is the damage variable D. Under uniaxial compression, the constitutive relation of concrete material after damage can be expressed as [31]
(5)σ=Eε(1−D)
where *σ* is stress, *ε* is strain, *E* is elastic modulus, and *D* is the damage variable. The damage variable is obtained by the transformation of Equation (5):(6)$$D=1-\frac{\sigma }{E\varepsilon }$$

The strain *ε* can be expressed as
(7)$$\varepsilon =\varepsilon_{e}+\varepsilon_{p}$$
where $\varepsilon_{e}$ is the elastic strain and $\varepsilon_{p}$ is the plastic strain. The elastic strain $\varepsilon_{e}$ can be derived from stress σ.
(8)$$\varepsilon_{e}=\frac{\sigma }{E}$$

The plastic strain $\varepsilon_{p}$ is expressed as
(9)$$\varepsilon_{p}=\varepsilon -\frac{\sigma }{E}$$

The relationship between the damage variable and the plastic strain under different loading speeds (strain rates) is shown in Figure 11. The damage variable *D* is calculated by calculating the elastic modulus *E* according to the linear elastic segment of the stress–strain curve and then substituting the stress and strain into Equation (6).

With the increase of the impact velocity, the crack propagation inside the specimen is hindered, and the evolution speed of the damage variable decreases, and under the same plastic strain, the damage variable corresponding to the high strain rate is lowered. When the stress of the concrete material is 70% of its peak value, it will enter the plastic stage [31]. According to the test results, when the ultra-early-strength cement-based material loses its bearing capacity, the corresponding damage variable is between 0.8 and 0.9, approximately.

According to the result of the analysis, the damage variable is related to strain rate $\dot{\varepsilon }$ and plastic strain $\varepsilon_{p}$. On this basis, the average value of the three damage variables under different strain rates was fitted. The comparison result of the fitting is shown in Figure 12. It can be seen from the figure that both are in a high degree of agreement. The fitting formula is as follows:(10)$$D=0.363\times \ln(1+A\varepsilon_{p})\;A=2437*\sin(0.054*\dot{\varepsilon }+1.39)\;7.5s^{-1}\leq \dot{\varepsilon }\leq 23.5s^{-1}$$

From the average value of the stress–strain curve of three repeated tests at different strain rates, the elastic modulus *E* (GPa) of the ultra-early-strength material at different strain rates is obtained. The relationship between elastic modulus *E* and strain rate is shown in Figure 13, and R^2^ is 0.98. By fitting the elastic modulus *E* at different strain rates, it is found that the relationship between the strain rate and the elastic modulus can be expressed as:(11)$$E=3.297\dot{\varepsilon }+33.53\;7.5s^{-1}\leq \dot{\varepsilon }\leq 23.5s^{-1}$$

By substituting the evolution variable expressions (8) and (10) under different strain rates into Equation (9), the stress–strain expression of the ultra-early-strength cement-based material can be obtained. The comparison between the results calculated by Equation (9) and the measured stress–strain results under different strain rates is shown in Figure 14. It can be found from the figure that the stress–strain curve calculated from the damage variables is in good agreement with the test results. Therefore, the internal damage evolution process of ultra-early-strength cement-based materials can be characterized by the damage variables.

## 6. Conclusions

In order to explore the dynamic mechanical properties of UR50 ultra-early-strength cement materials, an experimental study on the dynamic mechanical properties of ultra-early-strength cement-based materials at high strain rates was carried out using a large-diameter SHPB. The macroscopic failure morphology, dynamic stress–strain curve, and the relationship between dynamic compressive strength and strain rate of specimens under different strain rates (impact velocity) were obtained. According to the experimental results, the main conclusions are as follows:(1)Under different loading conditions, different types of copper sheets were selected as the shaper to eliminate the dispersion effect. Under the same impact velocity, the stress–strain curves of the three specimens are in good agreement, which ensures the validity and reliability of the experimental results;(2)The compression brittleness of ultra-early-strength cement-based specimens is relatively large, and the failure mode is a mainly brittle fracture, and with the increase of loading speed, the failure mode of the specimens gradually transited from larger fragments to small fragments, with an eventual large amount of powder;(3)The dynamic compressive strength of ultra-early-strength cement-based materials increases with the increase of the strain rate, which has an obvious strain rate strengthening effect. Fitting determines the relationship curve between the dynamic increase factor, DIF, and the strain rate. It has a linear relationship with the logarithm of the strain rate. The higher the strain rate, the larger the DIF, indicating that it has the advantage of impact-resistant mechanical properties;(4)The concept of absorption density is introduced to facilitate a better understanding of the toughness of ultra-early-strength cement-based materials. As the impact velocity increases, the peak stress rises, the energy absorption density value increases, and its growth rate accelerates. The peak strain at an impact velocity of 15 m/s is lower than that of low-speed impact (5 m/s and 10 m/s);(5)Based on the theory and method of continuum mechanics, the evolution process of the damage variables of ultra-early-strength cement-based materials was analyzed from a macro perspective. The damage variable equations at different strain rates were fitted according to the test results and based on the stress–strain curve, and the rationality of the damage evolution process was proved. With the increase in strain rate, the internal crack propagation of the specimen is hindered, and the increase rate of the damage variable decreases. Under the same plastic strain, the damage variable of the specimen under the high strain rate is relatively small.

In general, UR50 ultra-early-strength cement-based materials are more brittle in shock compression and will undergo an overall fracture at low strain rates. The dynamic compressive strength increases with the increase of the strain rate and has an obvious strain rate strengthening effect. However, further research on UR50 ultra-high early-strength concrete (UHESC) under SHPB-impact tests should be conducted. Additional research is also needed for the UR50 concrete base with high-speed impact to explore impact performance. The results of this research will further the development of dynamic material simulation methods and material models.

## Figures and Tables

**Figure 1 materials-15-06154-f001:**
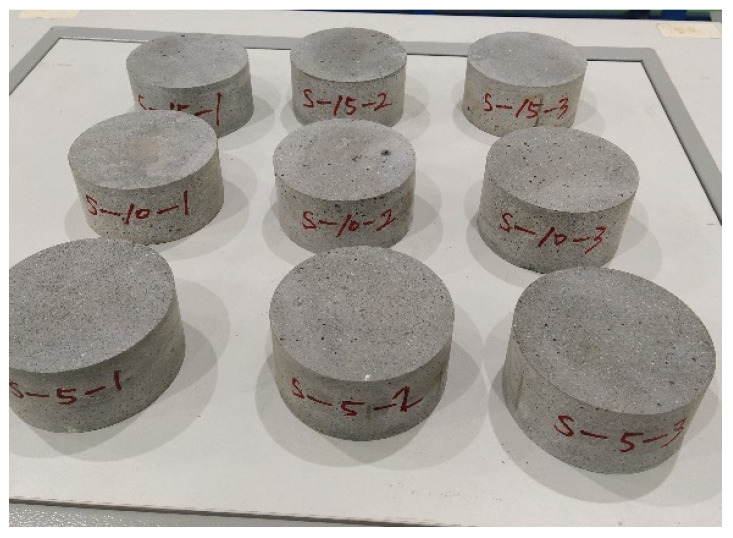
Physical image of processed specimens.

**Figure 2 materials-15-06154-f002:**
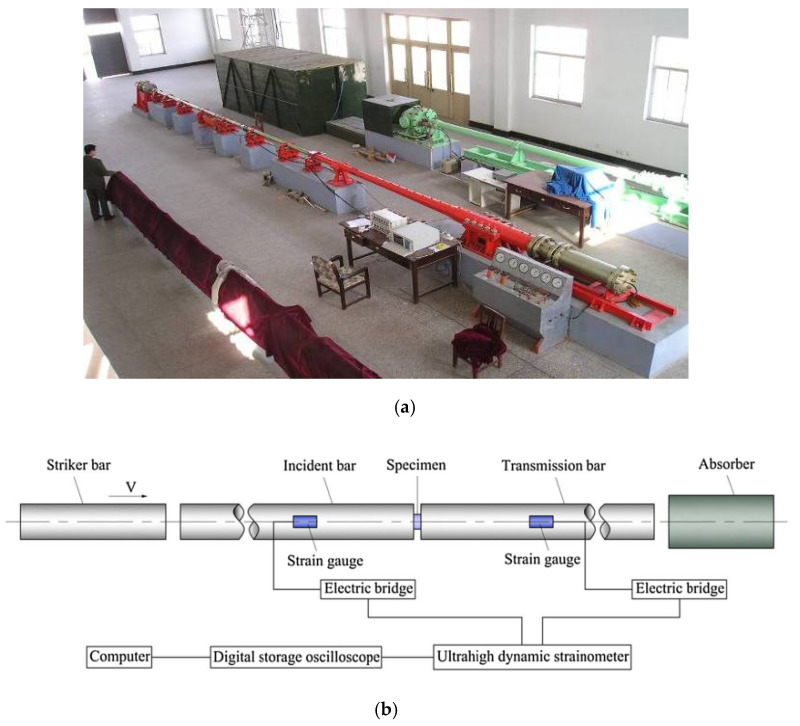
An SHPB device separated by 100 mm. (**a**) Physical map of the SHPB device. (**b**) Schematic diagram of the SHPB device.

**Figure 3 materials-15-06154-f003:**
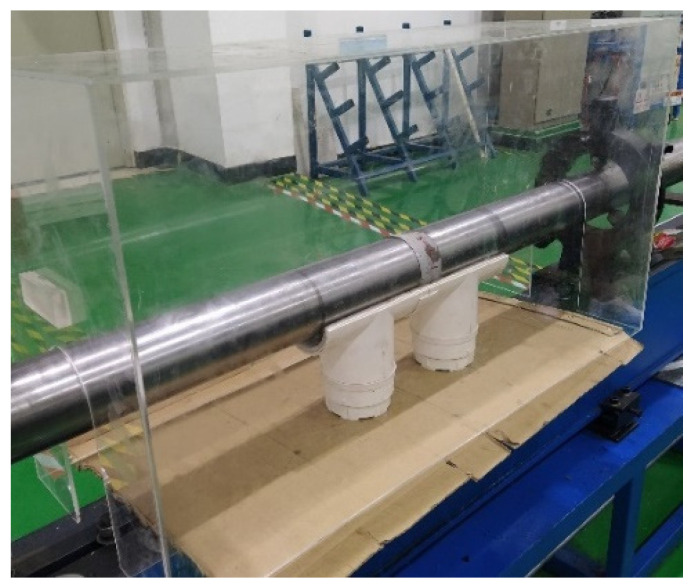
Physical photos of the test process.

**Figure 4 materials-15-06154-f004:**
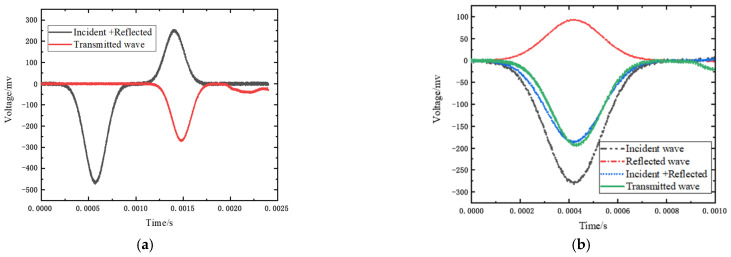
Original stress wave. (**a**) SHPB test collection wave. (**b**) The three waves graph.

**Figure 5 materials-15-06154-f005:**
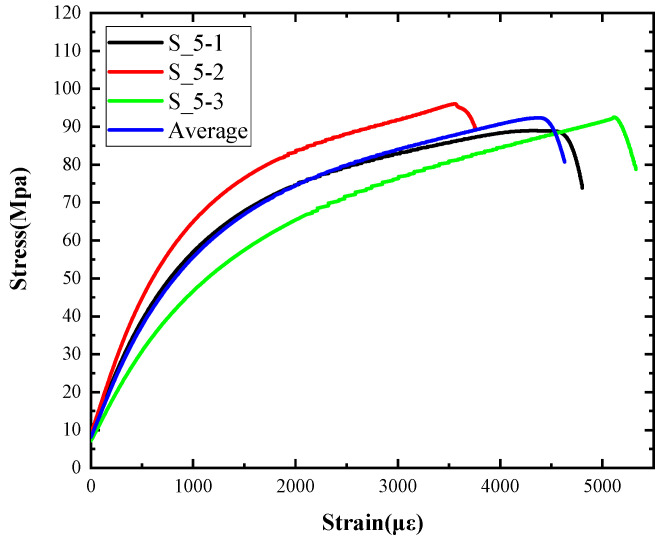
Stress–strain curve of 5 m/s.

**Figure 6 materials-15-06154-f006:**
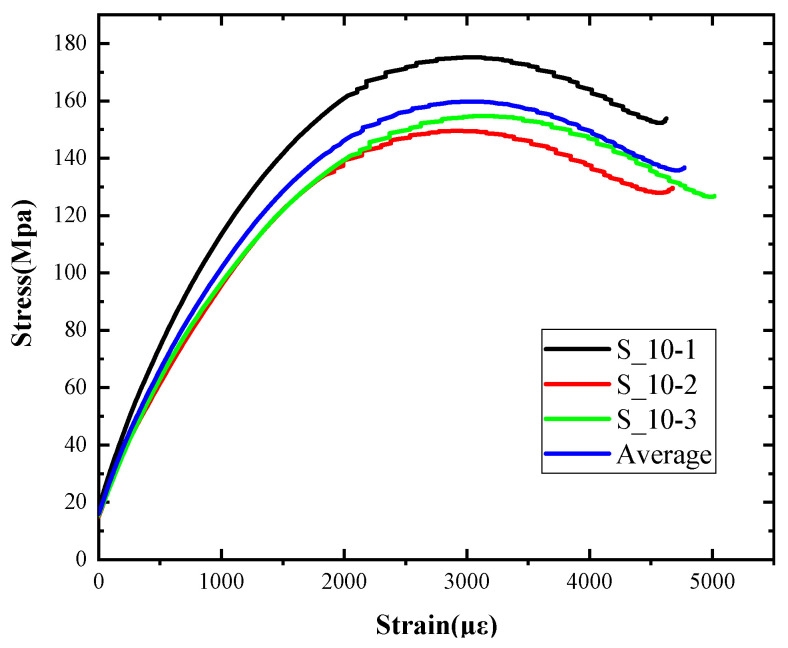
Stress–strain curve of 10 m/s.

**Figure 7 materials-15-06154-f007:**
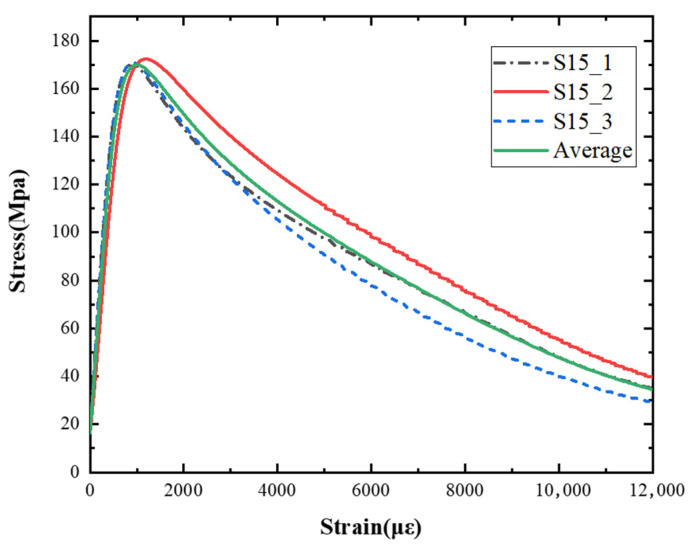
Stress–strain curve of 15 m/s.

**Figure 8 materials-15-06154-f008:**
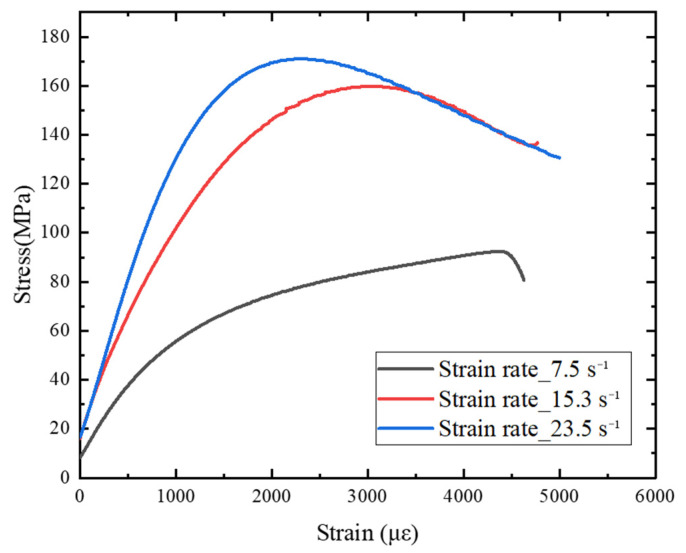
Dynamic stress–strain curve at different strain rates.

**Figure 9 materials-15-06154-f009:**
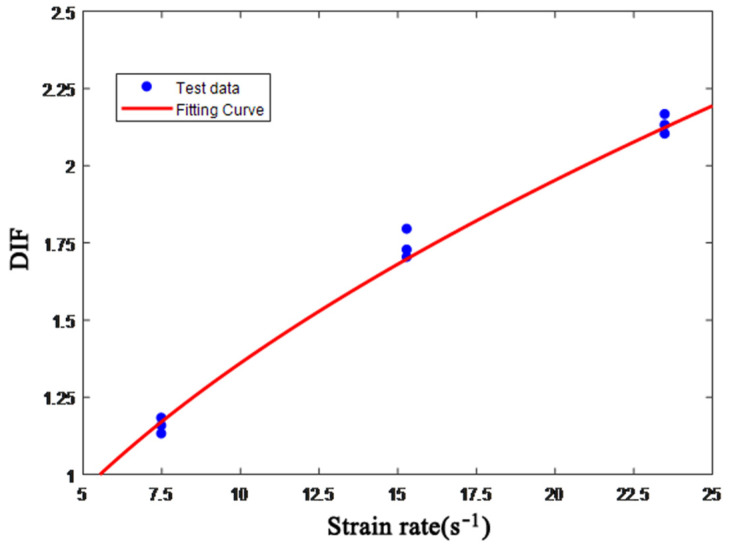
Relationship between DIF and strain rate.

**Figure 10 materials-15-06154-f010:**
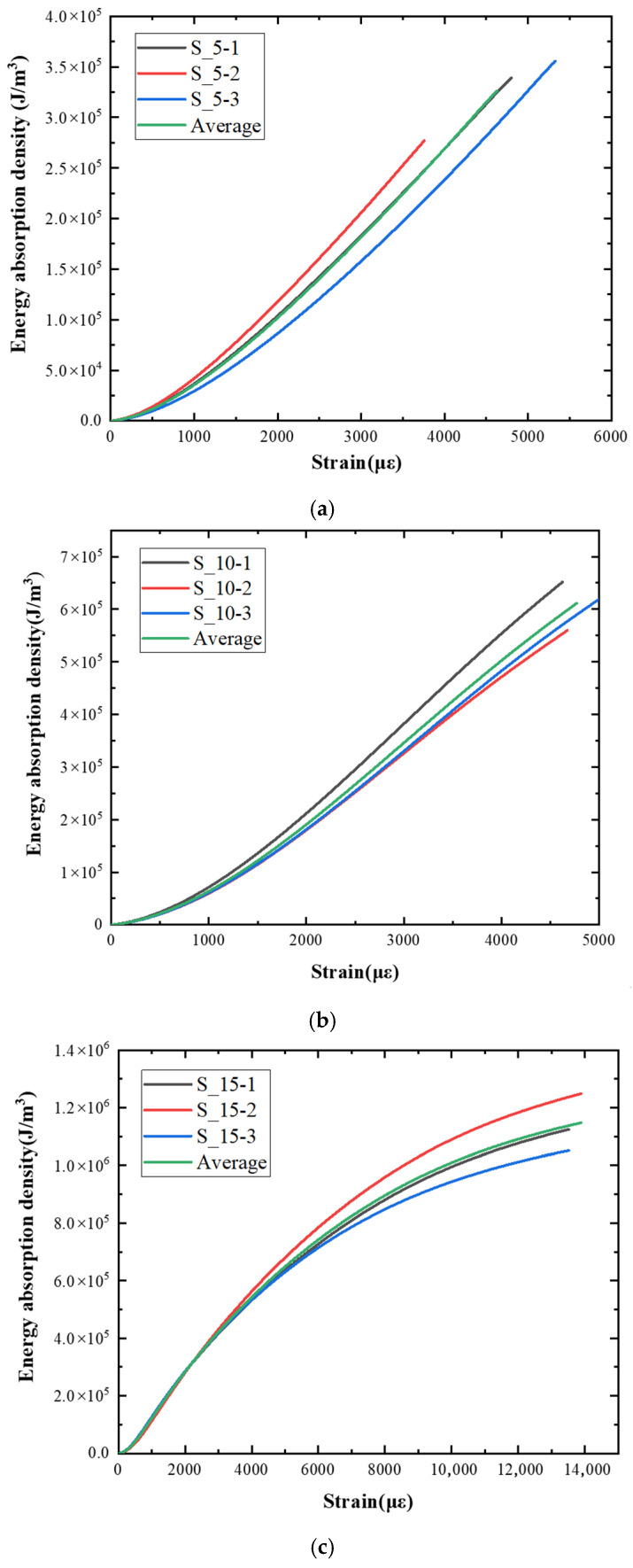
Energy absorption density–strain curve. (**a**) Impact velocity of 5 m/s. (**b**) Impact velocity of 10 m/s. (**c**) Impact velocity of 15 m/s. (**d**) Average value.

**Figure 11 materials-15-06154-f011:**
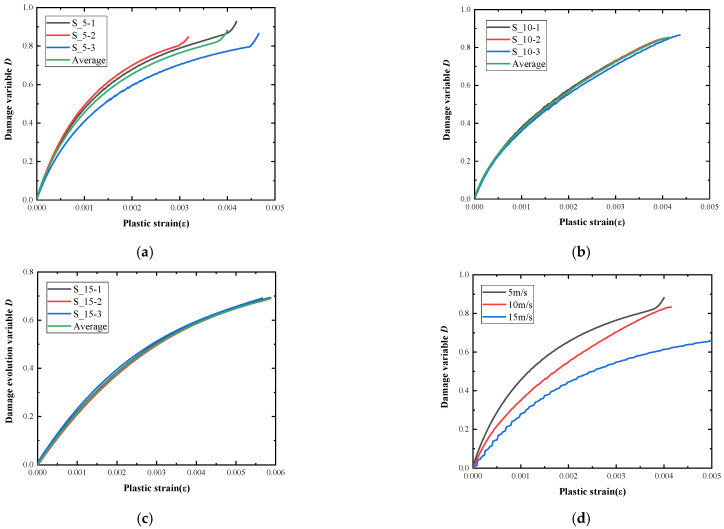
Damage evolution variable. (**a**) Impact velocity of 5 m/s. (**b**) Impact velocity of 10 m/s. (**c**) Impact velocity of 15 m/s. (**d**) Average value.

**Figure 12 materials-15-06154-f012:**
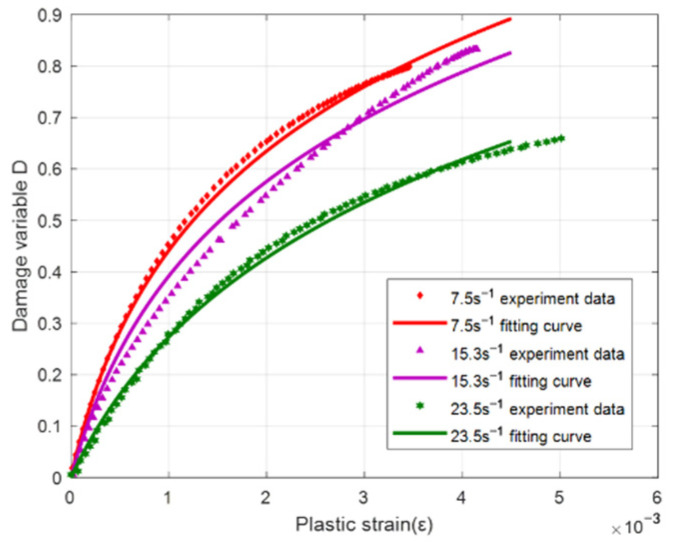
Comparison of damage variable fitting results.

**Figure 13 materials-15-06154-f013:**
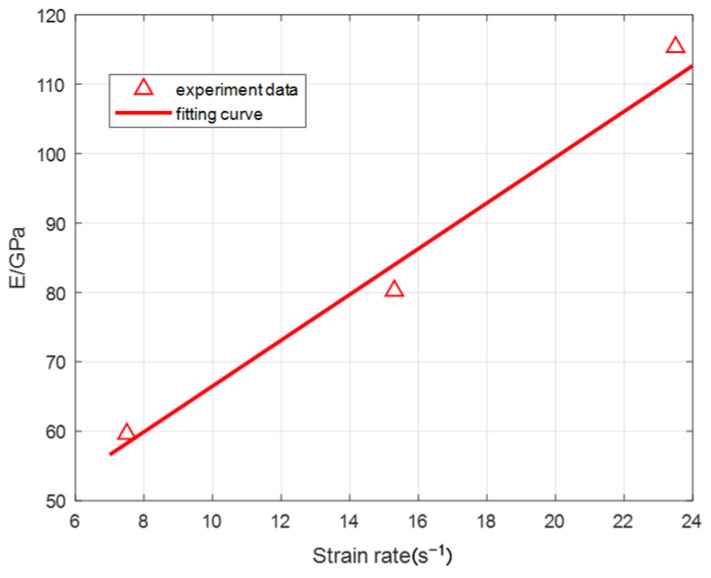
The relationship between elastic modulus and strain rate.

**Figure 14 materials-15-06154-f014:**
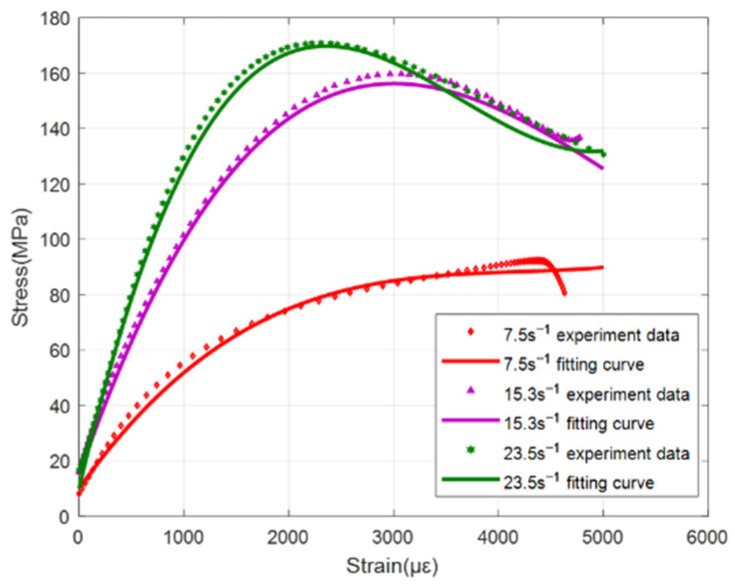
Comparison of fitting results with test results.

**Table 1 materials-15-06154-t001:** Mechanical properties of UR50.

Times	Compressive Strength (MPa)	Flexural Strength (MPa)
2 h	54.0	7.2
24 h	71.0	9.7
7 d	80.0	9.7
28 d	81.2	10.1

**Table 2 materials-15-06154-t002:** Experimental program.

Speeds	Specimen Number	Measured Size of Test Specimen
5 m/s	S_5-1	Φ101.50 × 48.84 mm
	S_5-2	Φ100.76 × 49.44 mm
	S_5-3	Φ101.08 × 51.44 mm
10 m/s	S_10-1	Φ101.40 × 49.24 mm
	S_10-2	Φ101.05 × 50.40 mm
	S_10-3	Φ101.40 × 50.54 mm
15 m/s	S_15-1	Φ101.32 × 50.12 mm
	S_15-2	Φ101.38 × 50.32 mm
	S_15-3	Φ101.42 × 51.10 mm

**Table 3 materials-15-06154-t003:** Test results at an impact velocity of 5 m/s.

Test No.	Before the Test	After the Test	Recycled
S_5-1	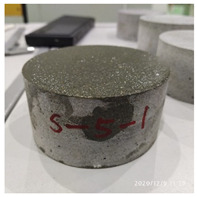	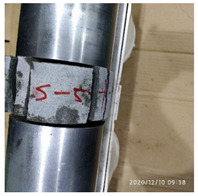	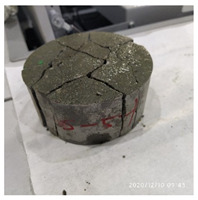
S_5-2	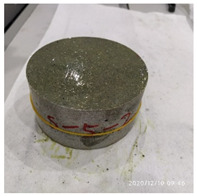	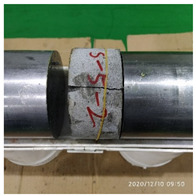	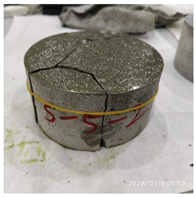
S_5-3	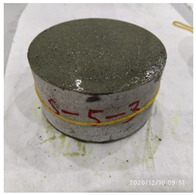	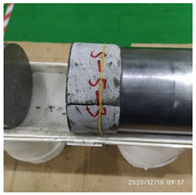	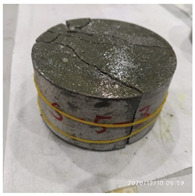

**Table 4 materials-15-06154-t004:** Test results at an impact velocity of 10 m/s.

Test No.	Before the Test	After the Test	Recycled
S_10-1	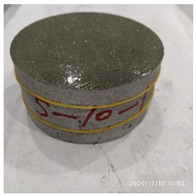	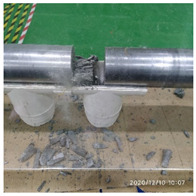	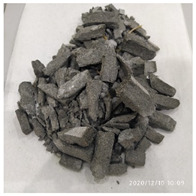
S_10-2	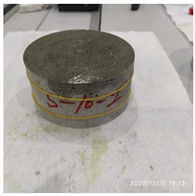	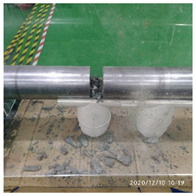	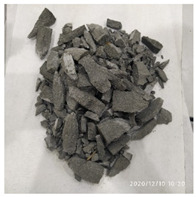
S_10-3	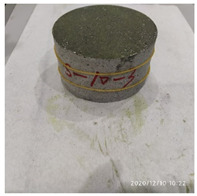	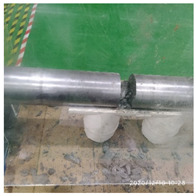	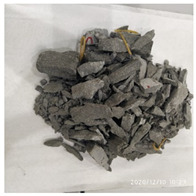

**Table 5 materials-15-06154-t005:** Test results at an impact velocity of 15 m/s.

Test No.	Before the Test	After the Test	Recycled
S_15-1	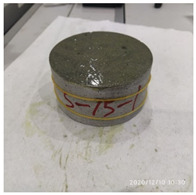	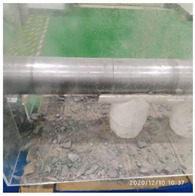	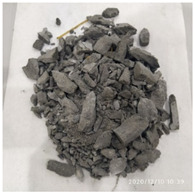
S_15-2	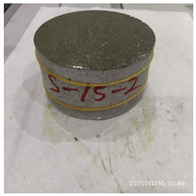	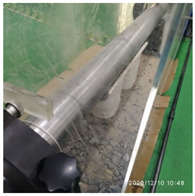	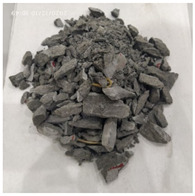
S_15-3	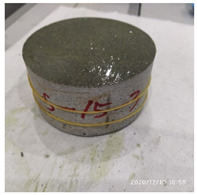	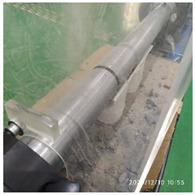	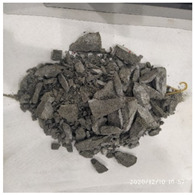

**Table 6 materials-15-06154-t006:** Dynamic compressive strength under different strain rates.

Strain Rates	Dynamic Compressive Strength (Mpa)		
7.5 (s^−1^)	93.954	95.982	91.836
15.3 (s^−1^)	146.194	149.262	144.668
23.5 (s^−1^)	173.013	175.845	170.697

## Data Availability

Not applicable.

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
