# Peer review of "Dynamic Compressive Mechanical Properties of UR50 Ultra-Early-Strength Cement-Based Concrete Material under High Strain Rate on SHPB Test"

_materials, 2022, doi:10.3390/ma15176154_

Round 1
Reviewer 1 Report
The article titled “Dynamic compressive mechanical properties of UR50 ultra-early-strength cement-based concrete material under high strain rate on SHPB test” presents technical information based on dynamic compressive properties of materials and numerical simulation studies. The test method and equipment are new. The results of experiment testing and numerical simulation are conformable. The manuscript should be major revised because there are many flaws that require improvement prior to acceptance. Please see attached the file.

Author Response
• I believe the impact test is the key experiment in the manuscript. It is suggested to add impact properties in the title. Response: Thank you for the valuable comment. Split hopkinson pressure bar (SHPB) is a device for testing the dynamic mechanical properties of materials. This paper studies the dynamic compressive properties of UR50 ultra-early-strength cement-based concrete material at high strain rates. Different velocities in this paper can put the material at different strain rates . Faster impact velocity puts the material at a high strain rate, which allows the material to achieve higher strength and better toughness. Thus, the title of this article is Dynamic compressive mechanical properties of UR50 ultra-early-strength cement-based concrete material under high strain rate on SHPB test.
- SHPB should be spelled out
Response: Thank you for the valuable comment. SHPB is split hopkinson pressure bar, which is a commonly used equipment for testing the dynamic mechanical properties of materials. Spelling it out in the title makes the title too long, and the full spelling will be reflected in the key words and text.
• Abstract: Line 13: Please sentence should be formally modified. Response: Thanks for pointing out the error. This sentence is modified as follows: Because of its ultra-high strength, ultra-high toughness, ultra-impact resistance and ultra-high durability, it has received great attention in the fields of protection engineering, but the dynamic mechanical properties of impact compression at high strain rates are known not well to people, and the dynamic compressive properties of materials are the basis for related numerical simulation studies.
• Line 16: split Hopkinson pressure bar (SHPB) should be the same as the term in Line 26 and 54. Response: Thanks for pointing out the error. This error has been corrected, and the full name of SHPB in the full text has been changed to split Hopkinson pressure bar. • UR50 should not be bold or it should be addressed in the keyword since it is a commercial code.Response: The reviewer's point of view is reasonable. UR50 is removed from the keywords in this article. Then the keyword is changed to ultra-early-strength concrete. • Line 46: Add more references from domestic and foreign researchers.Response: Thank you for the valuable comment. We have added the corresponding references from domestic and foreign researchers. But at the same time, because the introduction is a bit long, we have properly categorized some of the content.
- Line 110-112 : Carbon Fiber Reinforced Polymer (CFRP) was addressed twice
Response: Thanks for your valuable question. This sentence is modified as follows:(Xiong et al. 2019) studied the dynamic mechanical properties of Carbon Fiber Reinforced Polymer (CFRP) confined concrete at a high strain rate based on SHPB test device with a diameter of 155mm. The results show that CFRP confined concrete is not sensitive to the strain rate effect. • Line 172-173: “For the information on the microstructure and mix ratio of UR50, please refer to the literature(Wang et al. 2022).” The brief ingredients should be introduced here. After looking at the literature, the function of ultra-early-strength is from Sulphoaluminate cement, rather than “special cement”. Therefore, the description of Sulphoaluminate cement should be explained a bit. Please add some works describing its feature:Response: Thank you for the valuable comment. This sentence is modified as follows:Wang et al. [27] found that the microstructure was greatly improved compared to conventional high-strength concrete, the pores were eliminated, and the nano-microstructure was strengthened. The design of dry mixing and pre-dispersed low-proportion components greatly improves the strength and durability of concrete, and this method makes the microstructure of concrete more dense. • Line 165: Heading of Introduction is repeated with Line 32.Response: Thank you for the valuable comment. For the sake of title conflict, this article separates the UR50 ultra-early-strength concrete material as a title. • Figure 1: The coring process should not be present since it is a general methodResponse: The reviewer's point of view is reasonable. Since coring process is a general method, this article will not describe it here, and delete the relevant content. • Line 374:” European Concrete Council” (CEB) needs revised.Response: Thanks for your correction. This sentence is modified as follows:In this paper, the DIF model formula (4) of the concrete under the one-dimensional stress state approved by the Euro-International Committee for Concrete (the CEB) is used to fit the experimental data in Table 6. • Figure 10: use “DIF”Response: Thanks for your correction. The picture is modified as follows: • Figure 14: address the R2 of fitting curveResponse: Thank you for the valuable comment. The R2 of the fitted curve is 0.98. • Line 477-503: Please briefly conclude the concise findingsResponse: Thank you for the valuable comment. We add this paragraph at the end of the conclusion:In general, UR50 ultra-early-strength cement-based materials are more brittle in shock compression, and will undergo overall fracture at low strain rates. The dynamic compressive strength increases with the increase of the strain rate, and has obvious strain rate strengthening effect. • Line 504: What is the different between UR50 ultra-early-strength cement materials in this work and “UR50 ultra high early strength concrete (UHESC)”Response: Thanks for your valuable question. Ultra-high-early-strength concrete (UHESC) is a material developed based on the requirements of rapid repair and rapid construction. Its main function is that it can reach maturity in a short time compared to ordinary concrete. Ultra high early strength concrete is a large category, which contains UR50 ultra-early-strength cement materials.

Reviewer 2 Report
this is an interesting paper dealing with the cement-based concrete material, offering a series of basic data on high strain rate mechanical properties. It's well organized and structured. However, there are some issues needed to be clarified before it can be accepted for publication in Materials:
Introduction part is too long, it is better to reduce introduction part to the 3 paragraphs (and combine “Introduction to UR50” section in the Introduction)
Section 2.3 and 2.5 principle about SHPB could be removed, SHPB is a traditional testing method for study the material properties in the high strain rate and lots of books and articles published about the calibration and testing with SHPB, so it is not necessary to repeat here.
Line 205 incident waveguide bar, transmission waveguide bar should change to the incident bar and transmission bar.
Line 211 impact bar should change to striker bar.
Why did authors use two striker bars (500 and 800)?
As seen in the Fig.4 authors used pulse shaper, it would be great to explain it in the methodology part.
Test result should be presented based on the strain rate not impact velocity, please rearrange it.
It is better to rename the section 4 to discussion.
Author Response
1. Introduction part is too long, it is better to reduce introduction part to the 3 paragraphs (and combine “Introduction to UR50” section in the Introduction)Response: Thank you for your suggestion. The introductory part is indeed a bit long, and we have removed some unnecessary content. At the same time, because we need to better describe the properties of UR50 ultra-early-strength concrete material, we still keep the chapter on material properties, hoping to be understood. 2. Section 2.3 and 2.5 principle about SHPB could be removed, SHPB is a traditional testing method for study the material properties in the high strain rate and lots of books and articles published about the calibration and testing with SHPB, so it is not necessary to repeat here.Response: Thank you for your suggestion. Because SHPB is a traditional testing method for study the material properties in the high strain rate and lots of books and articles published about the calibration and testing with SHPB, we delete the principle part. For the calibration part of SHPB, we still keep it, the main reason is that it can describe the operation part of the test in more detail, so that readers can better understand our test. And in this part we also introduce the size and principle of the pulse shaper. 3. Line 205 incident waveguide bar, transmission waveguide bar should change to the incident bar and transmission bar.Response: Thank you for the valuable comment. This section is modified as follows:The impact compression test equipment is SHPB of Engineering Protection Research Institute, the test device is mainly composed of an operating console, launching device, impact bar(bullet), speed measuring device, incident bar, transmission bar, support, absorption bar, buffer device and measuring device, etc. 4. Line 211 impact bar should change to striker bar.Response: Thanks for pointing out the error. This sentence is modified as follows:The length of transmission bar is 2500mm, and the length of the striker bar is 500mm and 800mm respectively. 5. Why did authors use two striker bars (500 and 800)?Response: Thank you for the valuable comment. The reason why two different lengths of striker bars are used in this paper is that the mass of the striker bars of different lengths is different. At the same acceleration distance and the same air pressure, the 500mm striker bar gets more velocity than the 800mm striker bar. The variable tested in this paper is speed. For the safety of the test and the convenience of operation, we often use a 500mm striker bar to obtain a higher speed, and use an 800mm striker bar when the speed requirement is not high. 6. As seen in the Fig.4 authors used pulse shaper, it would be great to explain it in the methodology part.Response: Thank you for the valuable comment. The method and principle of the pulse shaper are explained in detail in the calibration of SHPB. 7. Test result should be presented based on the strain rate not impact velocity, please rearrange it.Response: Thank you for the valuable comment. Test results have been changed to be presented based on strain rate. 8. It is better to rename the section 4 to discussion.Response: Thank you for the valuable comment. This part of the content is really good with discussion as the title. This article has changed the title of Section 4 to discussion.

Reviewer 3 Report
1) First of all please prepare the final version of the manuscript, because at this moment I have red sentences and I do not know whether it is important or not.
2)Why do the authors of the concrete early strength test only after 28 days, what about the tests after 2 hours, 24 hours, and 7 days?
3) Was the prepared concrete reinforced?
4)In the research methodology, please add a subsection on the preparation of concrete mix and the results of its basic properties, i.e. consistency, air content, or pH
5)Were the samples before the dynamic test tested for bending and compression by commonly used methods?
6)Did the authors do any measurement error statistics? I know that there were only three samples tested, but for this case, there are also methods for determining the measurement uncertainty.
7)Abstract needs to modify: the abstract should contain Objectives, Methods/Analysis, Findings, and Novelty /Improvement.
8)More explanation is needed for where there is a research gap and what the goals of the research are. The research gap and the goals of the research are not explained in detail which leads to the reader missing the significance of the research.
9) Please add a sentence or two to clearly recap how your study differs from what has already been done in literature to ascertain the contributions more strongly.
10) For readers to quickly catch your contribution, it would be better to highlight major difficulties and challenges, and your original achievements to overcome them, in a clearer way in the abstract and introduction.
11) Some key parameters are not mentioned. The rationale for the choice of the particular set of parameters should be explained in more detail. Have the authors experimented with other sets of values? What are the sensitivities of these parameters on the results?
Author Response
1. Why do the authors of the concrete early strength test only after 28 days, what about the tests after 2 hours, 24 hours, and 7 days? Response: Thank you for the valuable comment. Standard maintenance is based on 28 days. At other times, such as 2 hours, 4 hours, and 7 days, the compressive strength of the material has not yet stabilized. Furthermore, this paper mainly studies the dynamic compressive mechanical properties of UR50 ultra-early-strength cement-based concrete material under high strain rate. This test after 2 hours, 24 hours and 7 days is a good direction certainly, which can be used as a direction for future research. 2. Was the prepared concrete reinforced? Response: Thanks for your valuable question. The prepared concrete was reinforced. During the test, we fixed the test piece with petroleum jelly and a bracket, which ensured that the test piece would not move during the test phase. At the same time, we also put a cover on the outside of the equipment to prevent debris from flying and ensure the accuracy and safety of the test.3. In the research methodology, please add a subsection on the preparation of concrete mix and the results of its basic properties, i.e. consistency, air content, or pHResponse: Thank you for the valuable comment. In this chapter of UR50 ultra-early-strength concrete material, we detail the preparation process of the material and the mechanical properties of the material at different times. 4. Were the samples before the dynamic test tested for bending and compression by commonly used methods?Response: Thanks for your valuable question. Sorry, the samples have not been tested in bending and compression by commonly used methods before the dynamic test. The main purpose of this paper is to study the dynamic compressive properties of materials and to lay the foundation for numerical simulation studies. 5. Did the authors do any measurement error statistics? I know that there were only three samples tested, but for this case, there are also methods for determining the measurement uncertainty.Response: Thanks for your valuable question. Sorry, we cannot do many test pieces due to test funding. Regarding the statistical method of measurement error you mentioned, we think this is a good point, and we can use this method in the following experimental studies. 6. Abstract needs to modify: the abstract should contain Objectives, Methods/Analysis, Findings, and Novelty /Improvement.Response: Thank you for the valuable comment. The abstract of this paper does have deficiencies. It is now revised as follows:UR50 ultra-early-strength cement-based self-compacting high-strength material is a special cement-based material. Compared with traditional high-strength concrete, its ultra-high strength, ultra-high toughness, ultra-impact resistance and ultra-high durability, it has received great attention in the fields of protection engineering, but the dynamic mechanical properties of impact compression at high strain rates are known not well to people, and the dynamic compressive properties of materials are the basis for related numerical simulation studies. In order to study its dynamic compressive mechanical properties, three sets of specimens with size of Φ100×50mm were designed and produced, large-diameter split Hopkinson pressure bar (SHPB) equipment with a diameter of 100mm were used to carry out impact tests with different speeds. The specimens were mainly brittle failure. With the increase of the impact speed, the failure mode of the specimens gradually transits from larger fragments to small fragments and a large amount of powder. The experimental results show that the ultra-early-strength cement-based material has greater impact compression brittleness, and overall rupture occurs at low strain rates. Its dynamic compressive strength increases with the increase of strain rates, and has obvious strain rate strengthening effect. According to the test results, the relationship curve between dynamic enhancement factor and strain rate is fitted. As the impact speed increases, the peak stress has been rising, the energy absorption density increases and its growth rate accelerates. Afterwards, based on the stress-strain curve, the damage variables under different strain rates were fitted, the results show that the increase of strain rate has a hindering effect on the increase of damage variables and the increase rate. 7. More explanation is needed for where there is a research gap and what the goals of the research are. The research gap and the goals of the research are not explained in detail which leads to the reader missing the significance of the research.Response: Thank you for the valuable comment. Ultra-early-strength cement-based materials, because of its excellent performance, have attracted great attention of airport engineering repair and protection engineering. Ultra-early-strength cement-based materials are usually subjected to strong dynamic load impact compression during service, like ammunition penetration and explosion. The dynamic compression performance of the material plays an important role in studying the dynamic mechanical response process of materials under penetration and explosion. However, there are few related researches on ultra-early-strength cement-based self-compacting high-strength materials. In contrast to previous studies, the dynamic compressive properties of the material were not investigated. In order to study the dynamic mechanical response process of materials under the action of penetrating explosion and lay the foundation for numerical simulation research, this paper conducts related research. 8. Please add a sentence or two to clearly recap how your study differs from what has already been done in literature to ascertain the contributions more strongly.Response: Thank you for the valuable comment. In order to clearly summarize how this study differs from the studies completed in the literature, we add the following sentences:In contrast to previous studies, the dynamic compressive properties of the material were not investigated. In order to study the dynamic mechanical response process of materials under the action of penetrating explosion and lay the foundation for numerical simulation research, this paper conducts related research. 9. For readers to quickly catch your contribution, it would be better to highlight major difficulties and challenges, and your original achievements to overcome them, in a clearer way in the abstract and introduction.Response: Thank you for the valuable comment. Based on your suggestion, we have revised the Abstract and Introduction. UR50 ultra-early-strength cement-based self-compacting high-strength material is a special cement-based material. Compared with traditional high-strength concrete, its ultra-high strength, ultra-high toughness, ultra-impact resistance and ultra-high durability, it has received great attention in the fields of protection engineering, but the dynamic mechanical properties of impact compression at high strain rates are known not well to people, and the dynamic compressive properties of materials are the basis for related numerical simulation studies. To this end, we designed relevant experiments to test its dynamic compression performance.10. Some key parameters are not mentioned. The rationale for the choice of the particular set of parameters should be explained in more detail. Have the authors experimented with other sets of values? What are the sensitivities of these parameters on the results?Response: Thanks for your valuable question. Sorry, due to trial funding, we didn’t experiment with other sets of values. We mainly make the specimens under different strain rates by using different impact velocities. After that, the macroscopic failure morphology, dynamic stress-strain curve and dynamic compressive strength of the specimen under different strain rates (impact speeds) were obtained.

Round 2
Reviewer 1 Report
The revised manuscript titled: "Dynamic compressive mechanical properties of UR50 ultra-early-strength cement-based concrete material under high strain rate on SHPB test" have good novel and sufficient information for publication.
Reviewer 3 Report
Thank you for all your responses. In my opinion, the manuscript is ready for the printing process.